# A Low-Cost IoT Sensors Network for Monitoring Three-Phase Induction Motor Mechanical Power Adopting an Indirect Measuring Method

**DOI:** 10.3390/s21030754

**Published:** 2021-01-23

**Authors:** Fabrizio Ciancetta, Edoardo Fiorucci, Antonio Ometto, Andrea Fioravanti, Simone Mari, Maria-Anna Segreto

**Affiliations:** 1Department of Industrial and Information Engineering and Economics (DIIIE), University of L’Aquila, Piazzale Ernesto Pontieri 1, Monteluco di Roio, 67100 L’Aquila, Italy; edoardo.fiorucci@univaq.it (E.F.); antonio.ometto@univaq.it (A.O.); andrea.fioravanti@graduate.univaq.it (A.F.); simone.mari@graduate.univaq.it (S.M.); 2LAERTE Laboratory (Italy), ENEA (Italian National Agency for New Technologies Energy and Sustainable Economic Development), Via Martiri di Monte Sole 4, 40129 Bologna, Italy; mariaanna.segreto@enea.it

**Keywords:** three-phase induction motor, indirect measurement, mechanical power, distributed measurement system, IoT

## Abstract

Three-phase induction motors are widely diffused in the industrial environment. Many times, the rated power of three-phase induction motors is not properly chosen causing incorrect operating conditions from an energetic point of view. Monitoring the mechanical dimension of a new motor is helpful, should an existing motor need to be replaced. This paper presents an IoT sensors network for monitoring the mechanical power produced by three-phase induction motors, adopting an indirect measuring method. The proposed technique can be easily adopted to monitor the mechanical power using only one line of current transducer, reducing the cost of the monitoring system. The proposed indirect measurement technique has been implemented on a low-cost IoT system, based on a Photon Particle SoC. The results show that the proposed IoT system can estimate the mechanical power with a relative error of within 8%.

## 1. Introduction

Nowadays, it is estimated that more than 70% of the electrical energy consumption in industrial environments is used to feed three-phase induction motors. The energy efficiency of a factory, therefore, depends primarily on the efficiency of its motors. The efficiency is affected by the actual mechanical power delivered by a factory’s motors, with respect to the motors’ rated power, and therefore it is crucial that appropriately sized motors are operated from an energetic point of view. On the one hand, a motor cannot be significantly oversized because it would operate at a low efficiency value. Installation and operating costs would also rise. On the other hand, a motor cannot be undersized because it would have a shorter operating life, caused by overheating [1]. Usually, a slightly oversized motor is used for safety reasons when the mechanical load is well known. Most of the time, in many small and medium companies, the mechanical load is not accurately estimated and the choice of the motor rated power is a difficult task. 

Useful information on the mechanical load can be obtained by monitoring a motor’s actual load power in a significant time interval in order to correctly identify under- or over-sized machines for replacement. Unfortunately, given that the load power measurement involves angular velocity and torque measurement, such a technique is not suitable for a large number of motors. Moreover, commercial data acquisition systems or data loggers for monitoring the mechanical power of many motors would be too expensive. Therefore, a low-cost slightly invasive monitoring system that can share the measured data over the Internet could be a good choice to monitor the operating conditions of three-phase induction motors installed in a company’s facility.

For these reasons, in this paper, we propose a low-cost IoT system that performs indirect measurements of mechanical power and publishes the data on a cloud system. Although new emerging technologies are spreading worldwide [2], the IoT technology has been chosen because it provides fast and effective access to data published on the web, and therefore is the most suitable for this application.

Mechanical power is evaluated by measuring a line current [3,4,5,6] and current measurements are carried out by using a low-cost transducer instead of a more sophisticated performing transducer [7,8,9]. These indirect measurements require a mathematical relationship between the mechanical power and a line current. Consequently, a model that is valid for a range of rated powers can be developed by starting from experimental measurements on motors with different rated powers.

The basic idea is to use the motor nameplate to identify the mechanical power vs. current characteristics, in order to estimate the mechanical power starting from the measurement of a line current without any other experimental analyses. 

Then, a low-cost IoT device is developed. The acquired signal is processed locally and the data are available on the Internet network in order to create a distributed and multipoint measurement system to have a clear scenario of the whole system [10,11].

The network architecture is based on a cloud system, which produces a separation between the IoT device and the user interface via web methods and web functions that allow data exchange. The data are displayed on an Android application to give much more freedom to the measurement network.

The validity of the line current–mechanical power relationship had been checked in a previous paper [3] by considering only the motors in the set used to find such a relationship. On the contrary, this paper refers to new experimental tests on other motors, out of the previous set. The new experimental tests are carried out, using the proposed measurement equipment, to evaluate the performance of both the proposed technique in terms of measurement errors and the hardware/software of the proposed and realized network architecture and IoT device.

## 2. The Proposed Technique

Induction motors can be considered to be systems that convert electrical power into mechanical power. Assuming sinusoidal voltage and current are at a steady state and the induction motor as a balanced three-phase load, the line current can be expressed [12,13] as:(1)$$I=\frac{P_{m}}{\sqrt{3}V_{p\text{-}p}\eta \left(P_{m}\right)\cos\phi \left(P_{m}\right)},$$
where:*I* is the line current;*P*_m_ is the mechanical power;*V_l_* is the line voltage;*η* is the efficiency;cos*ϕ* is the active factor.

Equation (1) can be used to evaluate the mechanical power, starting from the measurement of a line current [12,13,14], considering the voltage equal to the rated value. Unfortunately, the relation in Equation (1) it is not linear, since the functions *η(P*_m_*)* and cos*ϕ(P*_m_*)* depend on the mechanical power, as shown in [3] (Figure 1), where the typical trends of efficiency and active factor are plotted for different values of mechanical power.

The mechanical power as a function of line current *P*_m_ = *f*(*I*) can be approximated with a second order polynomial as follows:(2)$$P_{m}=aI^{2}+bI+c,$$
where *P*_m_ and *I* are expressed in p.u. values and the three coefficients (*a*, *b*, and *c*) depend on the parameter values of the motor equivalent circuit. 

High power induction motors are designed to obtain higher active factors and efficiencies. For this reason, the parameters of the equivalent circuits, as well as the coefficients *a*, *b,* and *c,* depend on the rated power [15,16,17,18]. Other parameters, such as the number of polar pairs, play a minor rule in the mechanical power–current relationship. The mechanical power of an induction motor can be estimated by measuring the line current by means of Equation (2). 

Thirteen three-phase induction motors with rated power from 1.1 to 75 kW have been tested to evaluate their coefficients *a*, *b,* and *c*. Five or more measurement points have been carried out for each motor [19,20,21] and data have been processed using the MatLAB environment to find the best fitting parabola, i.e., the coefficients *a*, *b,* and *c*. Thus, the measurement points and the best fitting parabola for the 13 three-phase induction motors considered are shown in [3], while the best fitting coefficients *a*, *b*, and *c* obtained for all the 13 motors are shown in [3] (Figure 4).

To give more flexibility to the proposed system, the results obtained for each single coefficient have been interpolated using the same function as follows:(3)$$y=\alpha +\beta \times \left(1-e^{\frac{x}{\gamma }}\right)+\delta \times x,$$
where *x* is the rated power expressed in kW and *y* represents the best fitting parameter (*a*, *b*, or *c*). The other constants of Equation (3) (*α*, *β*, *γ,* and *δ*) have been evaluated using the minimum square method. Their values and their trends vs. the motor rated power are reported in [3] (Table 2 and Figure 5).

## 3. Results Obtained from the Proposed Technique

In order to check the validity of the proposed method, the relative errors corresponding to five output powers (0.25, 0.50, 0.75, 1.00, and 1.25) p.u. have been evaluated using the following procedure:A motor among the 13 induction motors is selected.Given the value of the mechanical power ${\bar{P}}_{m}$ in one of the ranges (0.25, 0.50, 0.75, 1.00 and 1.25) p.u., the value of the corresponding line current ${\bar{I}}_{m}$ is evaluated using the original interpolated curve which accurately represents the current-mechanical power relationship.Adopting the proposed method, the mechanical power ${\overset{\bar{¯}}{P}}_{m}$ is estimated using the line current value ${\bar{I}}_{m}$(using the Equation (2)).The relative error between the indirectly estimated (${\bar{P}}_{m}$ ) and the directly measured (${\overset{\bar{¯}}{P}}_{m}$) power is evaluated.

The results obtained for each motor have been tabulated in Table 1. The obtained results show that the proposed method can be applied to estimate the mechanical power for three-phase induction motors ensuring a maximum relative error of 0.36%. At rated power, the maximum relative error is 0.02%. The proposed method, applied to three-phase induction motors with rated powers between 1.1 and 75 kW, can also be extended to other series of motors.

## 4. The Proposed IoT System 

Starting from the results obtained from the proposed method, a low-cost IoT system is developed to estimate the mechanical power of three-phase induction motors. The three main parts of the proposed system are shown in Figure 1, i.e., a low-cost split-core current transformer, a signal conditioning circuit that adapts the output of the current transducer to the input of the A/D converter embedded in the processing unit, and a Photon Particle-based network front-end and processing system. In the following sections, the main parts of the proposed system are described in more detail. 

### 4.1. Current Transducer

The low-cost current transducer selected for the proposed measurement system is the SCT-013-030 [22]. The transducer presents the following characteristics: (i) dielectric strength, 3 kV; (ii) operating range, 0 to 30 A; (iii) output range, 0 to 1 V; (iv) not linearity, ±3%, (v) mechanical dimensions, 13 × 13 mm; (vi) operating temperature, −25 to 70 °C., (as reported in Figure 2). The Fluke 6100 is used to calibrate the transducer, adopting increasing current values with steps ΔI = 0.5 A and decreasing currents with steps ΔI = 0.75 A. The results show that 1% is the maximum error introduced by the transducer.

### 4.2. Signal Conditioner

The signal conditioning circuit is the element that adapts the electrical characteristic of the transducer to the analog-to-digital converter (ADC) embedded in the processing unit. The SCT-013-030 transducer provides an output range of 0 to 1 V, for an input current of 0 to 30 A. Given that the input signal is sinusoidal, the transducer output values are in the range of −1 to 1 V. The ADC input range is 0 to 3.3 V. As shown in Figure 3, the signal conditioning circuit performs the following operations: (i) scales the output signal of the transducer from 2 to 3.3 V in order to use the entire ADC range and (ii) adds a constant voltage of 1.65 V to the transducer output signal to put the signal in the range of 0 to 3.3 V.

### 4.3. Processing and Network Front-End Unit

The Photon Particle device [23] (see Figure 4) is a system-on-chip (SoC) where a Broadcom BCM43362 Wi-Fi 802.11b/g/n and a STM32F205RGY6 120 MHz ARM Cortex M3 are present in the same chip. The main characteristics are 1 MB flash memory, 128 kB RAM, real-time operating system (FreeRTOS), and 18 mixed-signal GPIO. 

The built-in Wi-Fi module enables cloud communication via Internet access. Importantly, the oscillators that allow both chips to work are integrated within the same SoC. This additional degree of integration also simplifies the development of the board because the SoC does not require any other external components, but it is sufficient to power the SoC and both chips work. In addition, the SoC is certified for electromagnetic emission which does not allow installation restrictions in an ordinary environment. Finally, a built-in 3 × 12-bit A/D converter at 6 MSPS can acquire analog signals.

### 4.4. The Developed Board

The developed board is shown in Figure 5.

### 4.5. Signal Processing

The processing unit samples the signal from the condition circuit unit, executes the Root Mean Square (RMS) algorithm, and performs the proposed interpolation technique, starting from the rated power and the rated current. The RMS value of the line current can be calculated as:(4)$$RMS=K\times \sqrt{\frac{1}{M\times N}\sum_{i=0}^{\left(M\times N\right)-1}{\left[c\left(i\right)-c_{\mathrm{offset}}\right]}^{2}},$$
where:*K* is a constant which depends on the numbers of bit and on the range of analog-to-digital converter;*N* is the number of sampling points in one cycle;*M* is the number of acquired cycles;*c(i)* is the generic sampling code associated to the input signal;*c*_offset_ is the code which corresponds to the voltage offset introduced in order to put the input signal into the half of A/D range (0 to 3.3) V.

Introducing Δ*_c_*(*i*) = *c*(*i*) − *c*_offset_, the flowchart of the RMS implementation is reported in Figure 6. 

The ADC sampling frequency *f*_s_ is introduced and the time to evaluate the RMS values can be calculated as:(5)$$T_{\mathrm{acq}}=\frac{N_{\mathrm{points}}\times M_{\mathrm{cycles}}}{f_{s}}.$$

## 5. Network Architecture and Security Consideration 

Figure 7 shows the network architecture of the proposed system. Each IoT system was implemented using the Particle Proton device as the processing unit and network front-end, to access the Internet via a Wi-Fi connection. A local router creates the Wi-Fi network, where the IoT device can publish the processed data on a cloud system. If the number of induction motors, and therefore that of IoT devices, increases, it is possible to install more Wi-Fi routers for managing Internet traffic.

### 5.1. Access to Published Data via Cloud System

Particle provides a complete WEB platform for managing particle proton devices. It allows to control the status of the IoT system connected to the Internet and it also provides a complete integrated development environment (IDE) to download the firmware remotely directly via the WEB. Particle provides a complete cloud system to check both published data and published function, and to download data via API rest mechanism. 

The proposed IoT system uses this functionality to expose web functions and to upload the three-phase induction motor nameplate connected to the system (in terms of rated power and nominal current) in order to download the evaluated mechanical power. 

### 5.2. Network Security Consideration

The Particle Photon device adopts the http’s protocol for the Application Programming Interface (API) rest. The Secure Sockets Layer (SSL) creates a secure communication system between two endpoints, i.e., the IoT system and the client. The API rest web methods are available using an http query string approach, i.e., a formatted URL is able to access the WEB methods and WEB functions published by the remote device. The URL is organized with two main fields, IoT ID and access token. The IoT ID is 24 characters long and it is provided by Particle platform when the device is connected to the Web IDE. The access token is 40 characters long and it is associated with the developer. These two fields are related to the device, so the entire URL is not searchable by common web site and this functionality increases the intrinsic security of the system.

### 5.3. Client

An Android-based mobile client was developed to access the published data and configure the low-cost IoT system. Although a desktop-based solution is the most popular, a mobile Android client is very close to the idea of IoT, where the Thing can be connected anywhere to download data or send remote commands. 

In any case, the proposed network architecture allows a large degree of freedom, because the cloud system based on API rest web methods and web functions exposed by the Particle platform guarantees an abstraction between the IoT device and the web interface (see Figure 8). Since the application only needs to show mechanical power trends (data download) and sets the rated power of the three-phase induction motor (data load), the client is developed using the MIT App Invertor tool [24]. This platform offers an easy way to develop an Android application. As shown in Figure 9, the application has two screens as follows: The first screen plots the mechanical power over time and provides the actual data in a table and the second screen configures the application by setting the rated power and the rated current of the three-phase induction motor, uploading the data on the selected IoT device.

## 6. Measurement Setup 

The block diagram of the measurement setup is shown in Figure 10. A Yokogawa WT1800 power analyzer with motor evaluation function has been adopted for measuring electrical quantities, such as RMS voltage and current of each phase, as well as active input power and mechanical power output, to monitor the behavior of induction motors during tests. Torque and speed analog signals, supplied by the test bench, were also acquired. Motor enclosure temperature was monitored with a type J thermocouple and a Fluke 179 DMM. A photo of the measurement setup during testing is shown in Figure 11.

For the experimental verification of the proposed system, four three-phase induction motors of different rated power values were used. Their rated values are reported in Table 2.

## 7. Results 

Two tests were carried out as follows: (i) The first test was performed with the motor in thermal equilibrium, rapidly varying the load from 25% to 125% of the rated power, to keep the motor temperature constant (Figure 12). (ii) The second test was carried out with an induction motor loaded at rated power, during the thermal transient, with data acquisition performed in 4 °C steps near the thermal equilibrium (Figure 13).

## 8. Discussion and Conclusions

A low-cost monitoring system for three-phase induction motors that can share the measured data over the Internet is proposed. The aim of such a system is to estimate the output mechanical power of induction motors by sampling a line current and knowing the motor nameplate. The relationship between the rms value of the p.u. current and the p.u. mechanical power, based on theoretical considerations, was carried out experimentally by testing a set of 13 motors in a rated power range of 1.1 to 75 kW. A maximum relative error of 0.36% (0.02% at rated power) was obtained when motors in the tested set are considered.

Another four three-phase induction motors, out of the set used to tune the system, were considered to validate the proposed technique, and a maximum relative error of 8% at thermal steady state was found. Such a value is related to the motor with the lowest rated power (3 kW) and much smaller values correspond to motors with higher rated power. The following two aspects could justify this different behavior:First of all, low rated power motors have a larger standard deviation from the efficiency point of view; this affects the standard deviation of the values of the equivalent circuit parameters, and therefore the values of the coefficients *a*, *b,* and *c* used for the interpolation of the experimental points.The rate of variation of the coefficients *a*, *b,* and *c* is higher in the low rated power range.

In addition to these two aspects, the system can be improved by considering other motors parameters, such as:The pole pair number parameter, although it does not directly affect the relation between current and mechanical power, it can have an impact on the efficiency value. Generally speaking, two-pole motors have higher efficiency.The efficiency class of the machine, that obviously influences the relationship between current and mechanical power.

Further study should be carried out in order to take into account both pole pair numbers and efficiency classes of the motors.

Moreover, given that the current is sampled, the possibility of using the proposed system for condition monitoring to detect a fault in the operating conditions of the motor by means of the current THD factor should also be investigated. 

## Figures and Tables

**Figure 1 sensors-21-00754-f001:**
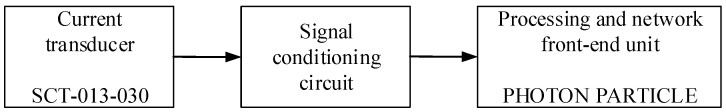
The main parts of the proposed IoT system.

**Figure 2 sensors-21-00754-f002:**
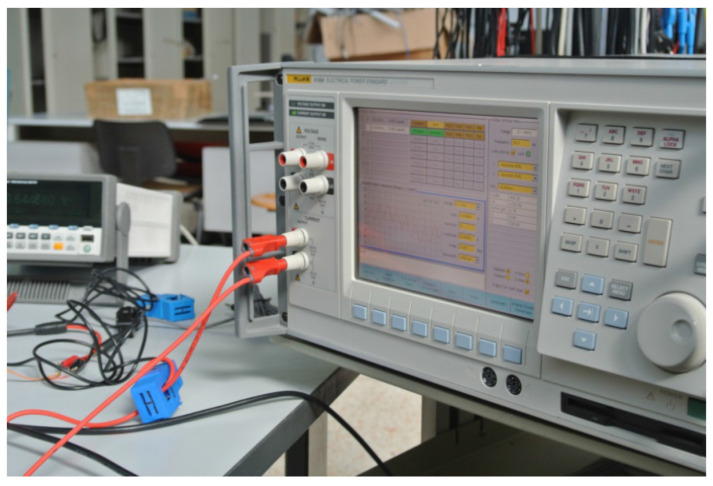
The SCT-013-030 during the calibration phase.

**Figure 3 sensors-21-00754-f003:**
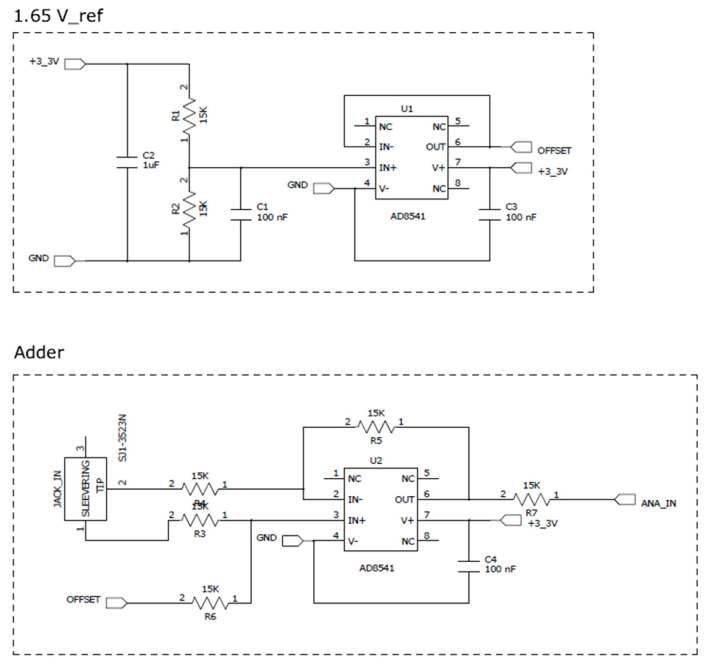
The schematic of signal conditioning circuit.

**Figure 4 sensors-21-00754-f004:**
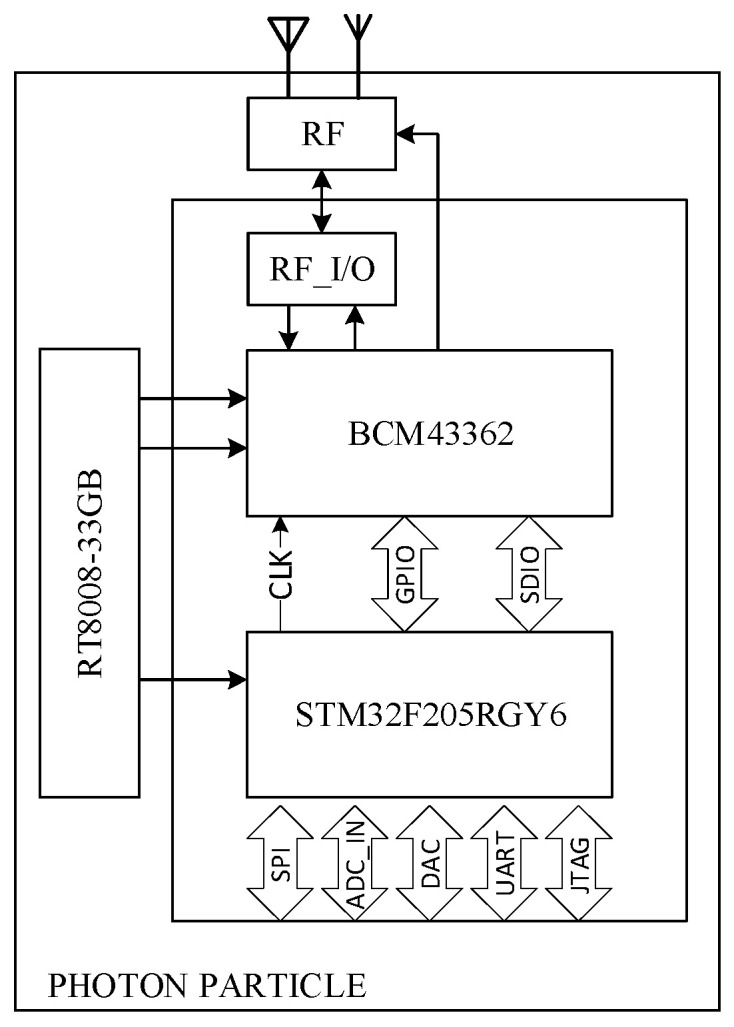
The Photon Particle detailed elements.

**Figure 5 sensors-21-00754-f005:**
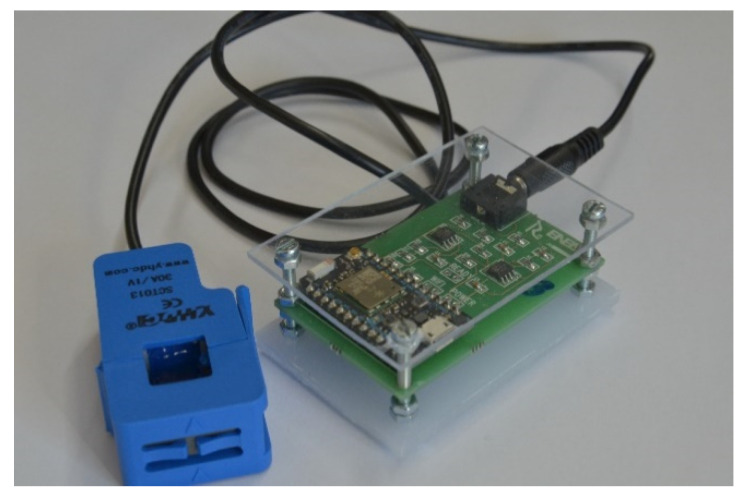
The developed board.

**Figure 6 sensors-21-00754-f006:**
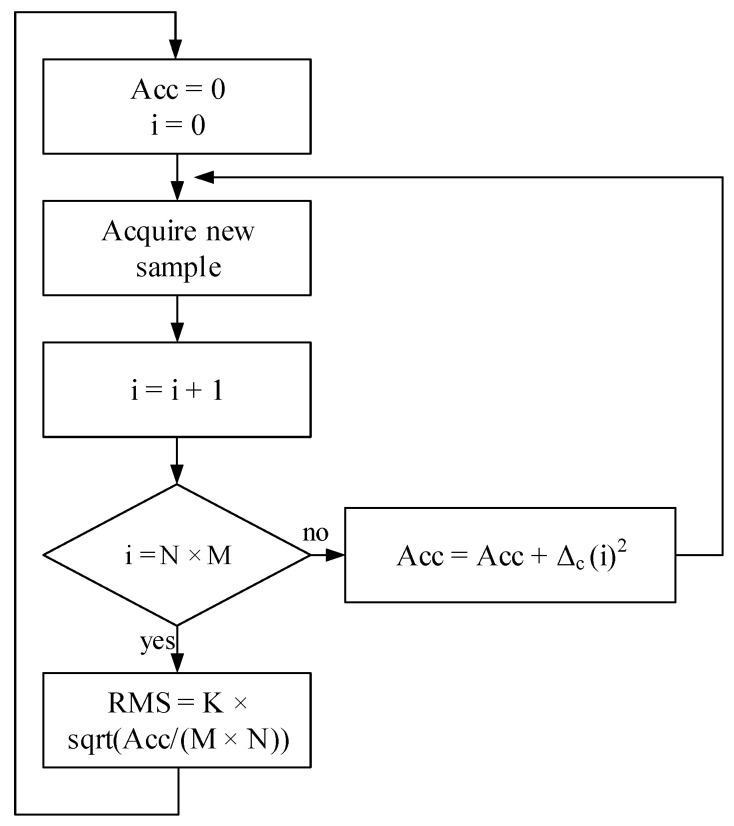
The flowchart of RMS implementation.

**Figure 7 sensors-21-00754-f007:**
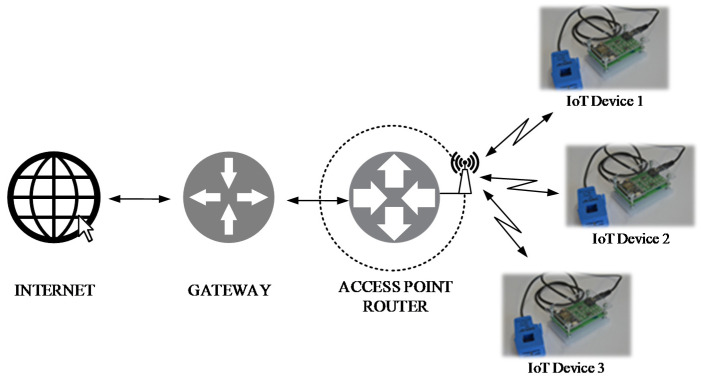
The network topology of the proposed IoT system.

**Figure 8 sensors-21-00754-f008:**
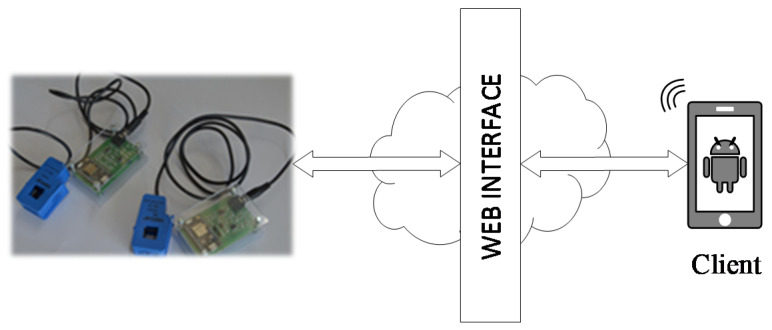
The web interface used to access the published data.

**Figure 9 sensors-21-00754-f009:**
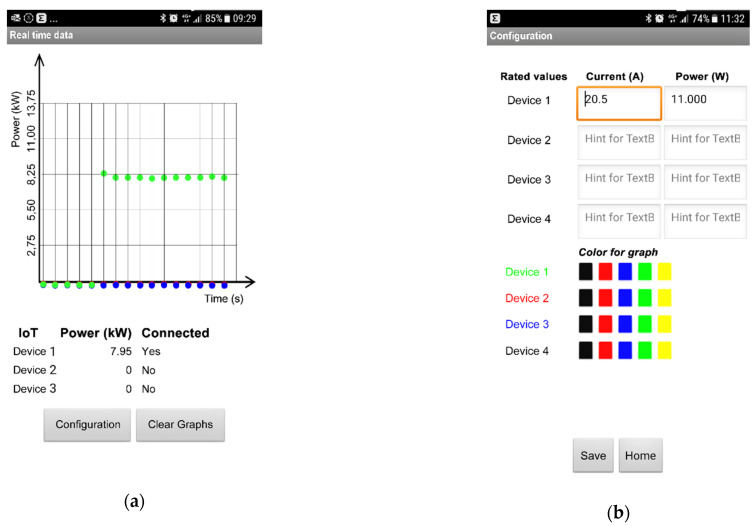
The screenshot of the client. (**a**) The main panel with mechanical power vs. time; (**b**) The configuration panel to upload the rated values for each device.

**Figure 10 sensors-21-00754-f010:**
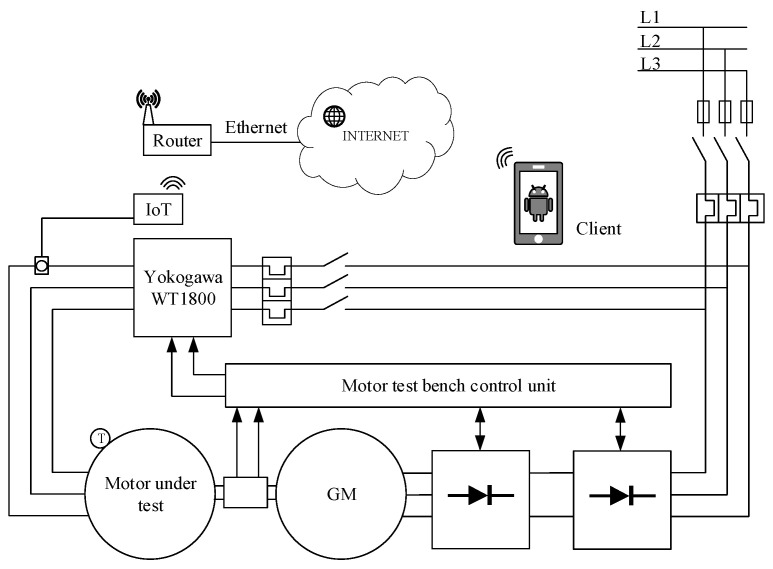
The block diagram of the test bench.

**Figure 11 sensors-21-00754-f011:**
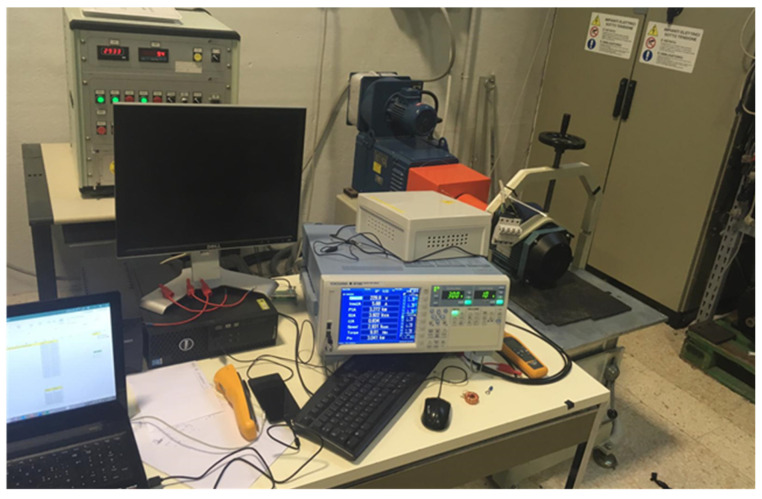
The measurement setup.

**Figure 12 sensors-21-00754-f012:**
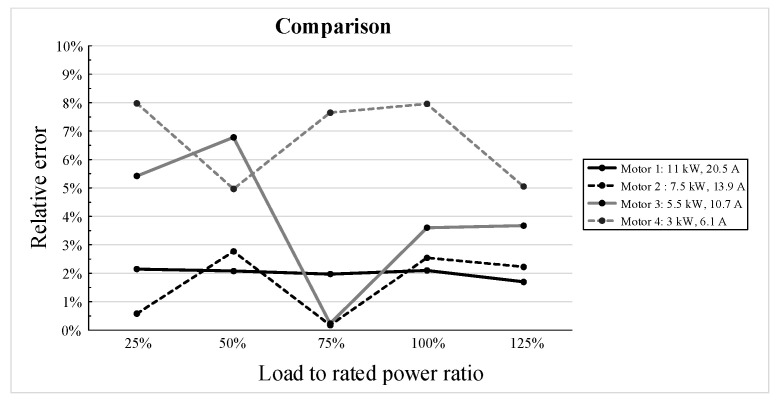
The relative error vs. load to rated power ratio for the four considered three-phase induction motors.

**Figure 13 sensors-21-00754-f013:**
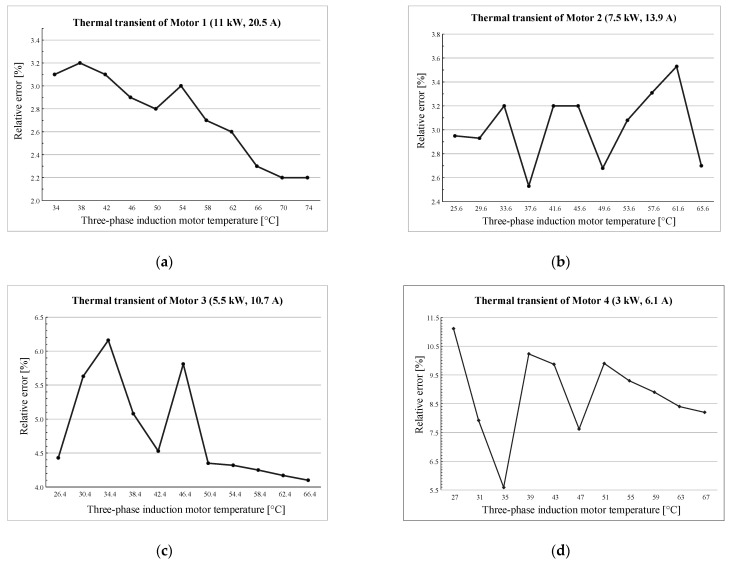
The relative error for thermal transient evaluated for the four considered three-phase induction motors. (**a**) Motor 1: 11.5 kW, 20.5 A; (**b**) Motor 2: 7.5 kW, 13.9 A; (**c**) Motor 3: 5.5 kW, 10.7 A; (**d**) Motor 4: 3 kW, 6.1 A.

**Table 1 sensors-21-00754-t001:** The error vs. motor mechanical power (in p.u.).

Motor	0.25	0.50	0.75	1.00	1.25
Motor 1 (1.1 kW)	0.298	0.181	0.074	−0.017	−0.076
Motor 2 (1.5 kW)	−0.024	−0.007	0.027	0.090	0.220
Motor 3 (1.5 kW)	0.472	0.248	0.031	−0.173	−0.349
Motor 4 (3 kW)	−0.023	0.027	0.041	0.006	−0.109
Motor 5 (3 kW)	−0.409	−0.351	−0.253	−0.09	0.199
Motor 6 (5.5 kW)	0.364	0.271	0.223	0.243	0.379
Motor 7 (7.5 kW)	−0.141	−0.060	0.010	0.061	0.083
Motor 8 (7.5 kW)	0.333	0.194	0.061	−0.061	−0.162
Motor 9 (11 kW)	−0.903	−0.597	−0.158	0.194	0.367
Motor 10 (22 kW)	0.174	0.129	0.073	−0.000	−0.099
Motor 11 (45 kW)	−0.226	−0.185	−0.152	−0.133	−0.132
Motor 12 (75 kW)	0.440	0.322	0.210	0.108	0.019
Motor 13 (75 kW)	−0.397	−0.280	−0.162	−0.045	0.071

**Table 2 sensors-21-00754-t002:** The rated values of three-phase induction motors used to validate the proposed system.

Motor	Power (kW)	Current (A)	Active Factor	Angular Speed (min^−1^)	Number of Poles
Motor 1	11	20.5	0.83	1465	4
Motor 2	7.5	13.9	0.90	2915	2
Motor 3	5.5	10.7	0.88	2900	2
Motor 4	3	6.1	0.80	1455	4

## Data Availability

The data presented in this study are available on request from the corresponding author.

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
