# Peer review of "A Low-Cost IoT Sensors Network for Monitoring Three-Phase Induction Motor Mechanical Power Adopting an Indirect Measuring Method"

_sensors, 2021, doi:10.3390/s21030754_

Round 1

Reviewer 1 Report

The paper is well written, quite thorough, and interesting. The analyzed dataset size of 13 motors is decent for a scientific paper.

Some minor comments:

- Section 2 references several figures of Reference [3]. For the reader's benefit, the figures should perhaps be replicated in the paper. If needed, the data from the original figures can be digitized by using e.g. WebPlotDigitizer, and the figures re-created with any changes or improvements deemed suitable. Obviously, the original must be adequately referenced, e.g. in the captions.

In equations, subscripts like 'points', 'cycles', etc. are usually not in italics. The authors could verify the recommended formatting rules for this journal, and revise if needed. For example, equations 2, 4, and 5 suffer from this problem.

Author Response

Response to Reviewer 1 Comments

Point 1: Section 2 references several figures of Reference [3]. For the reader's benefit, the figures should perhaps be replicated in the paper. If needed, the data from the original figures can be digitized by using e.g. WebPlotDigitizer, and the figures re-created with any changes or improvements deemed suitable. Obviously, the original must be adequately referenced, e.g. in the captions.

Response 1: Thanks for your comments.

Point 2: In equations, subscripts like 'points', 'cycles', etc. are usually not in italics. The authors could verify the recommended formatting rules for this journal, and revise if needed. For example, equations 2, 4, and 5 suffer from this problem.

Response 2: Thanks for your comments. The whole paper has been revised paying more attention on the formulas. We hope that everything is all right now.

Reviewer 2 Report

The paper explains the system developed by the authors. It's not rocket science, but seems well done and of some interest to the reader.

However, the presentation needs to be improved. There is a large number of writing mistakes in formulae, which make reading needlessly painful. Nobody would dare submitting running text in this state. Since writing correctly is no difficult task, there needs to be another review to check this problem is solved. Formulae must follow the usual and commonly accepted conventions for mathematical formulae and expressions (see, e.g., SI brochure (online, free of charge), NIST SP 811 (online, free of charge), ISO 80000). In particular, descriptive text (e.g. p-p, m, points) is to be written in upright (Roman) font. Quantity symbols shall be single-letter symbols (e.g. N, not "code"). Quantity symbols (e.g. I, n) shall be printed in italic type, also in the text. The conventional symbol for multiplication is either the half-height dot or the cross (Alt-0215), not x (l. 169) and not * (Fig. 6). The star * has a different meaning. Ranges need to be specified in a mathematically correct form, especially not 0-3.3 V (which is equal to -3.3 V, not but a range). The decimal marker should be the same throughout the document (either . or ,).

Line 16: The second sentence has no link to the rest of the abstract and seems to be misplaced. It is a claim, probably correct, but the abstract doesn't mention that the paper helps solving this problem. I have the impression that, rather than removing the sentence, another sentence with some explanation could help (e.g. adding a new sentence in l. 21 "Monitoring will help dimensioning a new motor, should the existing one need to be replaced").

Eq. (3): The coefficient e is not introduced. What is it? If it is meant to be e = 2.718..., please correct. Mathematical constants are to be set in upright (Roman) type. This does change the meaning.

Lines 155f: V is the unit symbol for volt. There is no special unit for RMS values or peak-to-peak voltage. So it is wrong to add modifiers to unit symbols, as discussed in all relevant documents (see above). Remove.

Line 162: There is to be a space between the numeric value and the unit symbol (except for angular unit symbols °, ', "). There should not be kelvin-bytes (symbol KB), but kilobytes (symbol kB). Correct.

Line 249: Even though rpm is often used, the conventional symbol is min^{-1}. Is the power factor really on the name plate? If so, spell out or introduce abbreviation PF properly. The standard symbol is lambda. I'm used to seeing cos phi (which is the active factor, not the power factor, see, e.g., IEC 60050 ( http://www.electropedia.org/iev/iev.nsf/display?openform&ievref=131-11-46 and http://www.electropedia.org/iev/iev.nsf/display?openform&ievref=131-11-49 ). Specify ranges correctly.

Figures 12: The axis labels need improvement. Figure 12 is probably wrong, because the axis labels specify the unit, but the unit is repeated in the tick marks. So you claim to give the load to rated power ratio from 0.0025 = 25 %×% to 0.0125 = 125 %×%. Which is not interesting.

Author Response

Response to Reviewer 2 Comments

Point 1: The paper explains the system developed by the authors. It's not rocket science, but seems well done and of some interest to the reader.

However, the presentation needs to be improved. There is a large number of writing mistakes in formulae, which make reading needlessly painful. Nobody would dare submitting running text in this state. Since writing correctly is no difficult task, there needs to be another review to check this problem is solved. Formulae must follow the usual and commonly accepted conventions for mathematical formulae and expressions (see, e.g., SI brochure (online, free of charge), NIST SP 811 (online, free of charge), ISO 80000). In particular, descriptive text (e.g. p-p, m, points) is to be written in upright (Roman) font. Quantity symbols shall be single-letter symbols (e.g. N, not "code"). Quantity symbols (e.g. I, n) shall be printed in italic type, also in the text. The conventional symbol for multiplication is either the half-height dot or the cross (Alt-0215), not x (l. 169) and not * (Fig. 6). The star * has a different meaning. Ranges need to be specified in a mathematically correct form, especially not 0-3.3 V (which is equal to -3.3 V, not but a range). The decimal marker should be the same throughout the document (either . or ,).

Response 1: Thanks for your comments. The whole paper has been revised paying more attention on the formulas and on the measurement units. We hope that everything is all right now.

Point 2: Line 16: The second sentence has no link to the rest of the abstract and seems to be misplaced. It is a claim, probably correct, but the abstract doesn't mention that the paper helps solving this problem. I have the impression that, rather than removing the sentence, another sentence with some explanation could help (e.g. adding a new sentence in l. 21 "Monitoring will help dimensioning a new motor, should the existing one need to be replaced").

Response 2: Thank you very much. We added the suggested sentence.

Point 3: Eq. (3): The coefficient e is not introduced. What is it? If it is meant to be e = 2.718..., please correct. Mathematical constants are to be set in upright (Roman) type. This does change the meaning.

Response 3: see Response 1

Point 4: Lines 155f: V is the unit symbol for volt. There is no special unit for RMS values or peak-to-peak voltage. So it is wrong to add modifiers to unit symbols, as discussed in all relevant documents (see above). Remove.

Response 4: see Response 1

Point 5: Line 162: There is to be a space between the numeric value and the unit symbol (except for angular unit symbols °, ', "). There should not be kelvin-bytes (symbol KB), but kilobytes (symbol kB). Correct.

Response 5: see Response 1

Point 6: Line 249: Even though rpm is often used, the conventional symbol is min^{-1}. Is the power factor really on the name plate? If so, spell out or introduce abbreviation PF properly. The standard symbol is lambda. I'm used to seeing cos phi (which is the active factor, not the power factor, see, e.g., IEC 60050 ( http://www.electropedia.org/iev/iev.nsf/display?openform&ievref=131-11-46 and http://www.electropedia.org/iev/iev.nsf/display?openform&ievref=131-11-49 ). Specify ranges correctly.

Response 6: see Response 1 We have changed power factor in active factor but given that we assume sinusoidal voltage and current, they have the same value.

Point 7: Figures 12: The axis labels need improvement. Figure 12 is probably wrong, because the axis labels specify the unit, but the unit is repeated in the tick marks. So you claim to give the load to rated power ratio from 0.0025 = 25 %×% to 0.0125 = 125 %×%. Which is not interesting.

Response 7: Thanks for your comment. We change that axis labels because the units are reported on the graph.

Reviewer 3 Report

A research article (manuscript ID: sensors-965829) entitled “A low-cost IoT sensors network for monitoring three-phase induction motor mechanical power adopting an indirect measuring method” by a collaborative research team from Italy was submitted as a letter to the MDPI Sensors Journal.

This presentation contains 13 pages including 13 figures, one table, and 25 references. In their paper, the authors have represented an IoT sensors network for the monitoring of mechanical power produced by three-phase induction motors, adopting an indirect measuring method. This work can be interesting for theoreticians and experimentalists as well as for graduate and postgraduate students.

The reviewer has looked through the paper and found that this presentation requires a major revision.

Firs of all, it is necessary to state that refs. [23, 24, 25] contain only internet pages that are in general, unpublished materials. I am not sure that the Journal can accept such references.

Second, the table should be on the same page but not on both pages 3 and 4. It should be used “dot” instead of “comma” for all the numbers present in the table.

There is no CONCLUSION that should be added.

Also, the article “the” must present at the beginning of all the titles for the figures and tables. For instance,

“Table 1. The error” instead of “Table 1. Error”;

“Figure 3. The schematic” instead of “Figure 3. Schematic”;

“Figure 4. The particle” instead of “Figure 4. Particle”;

“Figure 5. The developed board.” instead of “Figure 5. Developed board.”;

“Figure 6. The flowchart” instead of “Figure 6. Flowchart”;

“Figure 7. The network” instead of “Figure 7. Network”;

“Figure 8. The web” instead of “Figure 8. Web”;

“Figure 9. The screenshot” instead of “Figure 9. Screenshot”;

“Figure 10. The block” instead of “Figure 10. Block”;

“Figure 11. The measurement setup.” instead of “Figure 11. Measurement setup.”;

“Figure 12. The comparison” instead of “Figure 12. Comparison”;

“Figure 13. The relative error” instead of “Figure 2. Relative error”;

Also, in formulas (3), (4), and (5), the authors have not to use the sign “*” or another but they should use “×” or NOTHING.

The English language must be polished. The corresponding figures and formulas must be improved. The paper requires a major revision.

Author Response

Response to Reviewer 3 Comments

Point 1: Firs of all, it is necessary to state that refs. [23, 24, 25] contain only internet pages that are in general, unpublished materials. I am not sure that the Journal can accept such references.

Response 1: The Journal accepts references to website as reported in the Model Paper

Point 2: Second, the table should be on the same page but not on both pages 3 and 4. It should be used “dot” instead of “comma” for all the numbers present in the table.

Response 2: Thanks for your comment. The Table has been placed in page n.4 and all the “,” has been replaced with “.”.

Point 3: There is no CONCLUSION that should be added.

Response 3: The section 8 has been renamed in “Discussion and Conclusions”

Point 4: Also, the article “the” must present at the beginning of all the titles for the figures and tables. For instance,

“Table 1. The error” instead of “Table 1. Error”;

“Figure 3. The schematic” instead of “Figure 3. Schematic”;

“Figure 4. The particle” instead of “Figure 4. Particle”;

“Figure 5. The developed board.” instead of “Figure 5. Developed board.”;

“Figure 6. The flowchart” instead of “Figure 6. Flowchart”;

“Figure 7. The network” instead of “Figure 7. Network”;

“Figure 8. The web” instead of “Figure 8. Web”;

“Figure 9. The screenshot” instead of “Figure 9. Screenshot”;

“Figure 10. The block” instead of “Figure 10. Block”;

“Figure 11. The measurement setup.” instead of “Figure 11. Measurement setup.”;

“Figure 12. The comparison” instead of “Figure 12. Comparison”;

“Figure 13. The relative error” instead of “Figure 2. Relative error”;

Response 4: All the captions are changed

Point 5: Also, in formulas (3), (4), and (5), the authors have not to use the sign “*” or another but they should use “×” or NOTHING.

Response 5: Thanks for your comments. The whole paper has been revised paying more attention on the formulas and on the measurement units. We hope that everything is all right now.

Point 6: The English language must be polished. The corresponding figures and formulas must be improved. The paper requires a major revision.

Response 6: The whole paper has been revised.

Round 2

Reviewer 2 Report

Congratulations.

Reviewer 3 Report

A research article (revised manuscript ID: sensors-965829) entitled “A low-cost IoT sensors network for monitoring three-phase induction motor mechanical power adopting an indirect measuring method” by a collaborative research team from Italy was submitted as a letter to the MDPI Sensors Journal.

This presentation contains 13 pages including 13 figures, two tables, and 25 references. In their paper, the authors have represented an IoT sensors network for the monitoring of mechanical power produced by three-phase induction motors, adopting an indirect measuring method. This work can be interesting for theoreticians and experimentalists as well as for graduate and postgraduate students.

The authors have corrected many things in their paper. So, the reviewer has no arguments to reject this revised paper. This means that the paper can be published after some polishing in the English language.